## PERSPECTIVE

# Channelling protons out of the heart

Pawel Swietach[1] [ID] and Sanda Despa[2]

[1]*Department of Physiology, Anatomy & Genetics, Oxford, UK*

[2]*Department of Pharmacology and Nutritional Sciences, University of Kentucky, Lexington, Kentucky, USA*

Email: pawel.swietach@dpag.ox.ac.uk

Edited by: Bjorn Knollmann & Eleonora Grandi

Linked articles: This Perspective article highlights an article by Ma *et al.* To read this paper, visit https://doi.org/10.1113/JP282126.

The peer review history is available in the Supporting information section of this article (https://doi.org/10.1113/JP283250#support-information-section).

Acid-base carriers were among the first membrane transport processes to be investigated in the 1970s. Since the advent of methods for isolating cells, many studies of pH regulation used cardiac myocytes, ostensibly because of the clinical relevance of acidosis in ischaemia. Ma et al. (2022) present possibly the biggest paradigm shift in the field in 20 years by showing evidence for a role of HVCN1 voltage-gated $H^+$ channels in expunging acid from cells, at least in canine myocytes. Unlike $Na^+/H^+$ exchanger-1 (NHE1), the heart's most celebrated pH regulator, HVCN1 is electrogenic and not stoichiometrically coupled to $Na^+$ uptake. The authors argue that this novel $H^+$ extrusion pathway would be cardioprotective, as it avoids disrupting calcium signalling secondary to $Na^+$ overload – an issue with $Na^+$-dependent acid-extruders. The authors undertook extensive electrophysiological studies to verify their discovery. The fluxes attributed to HVCN1 are truly large; for example, dialysing cells with an acidic internal solution (pHi = 6.5) that normally triggers NHE1 activity in the order of tens of mM/min evoked an outward current of $\sim$1 pA/pF at 0 mV under near-physiological extracellular pH. This can be converted to a $H^+$ flux by considering the Faraday constant (96,500 A s/mol) and a representative ratio of membrane capacitance to cytosolic volume (Satoh et al., 1996) in mammalian myocytes (9 pF/pl). The calculated flux is 6 mmol/(l cytosol)/min, i.e. comparable to that produced by NHE1, bringing HVCN1 to the forefront of pH regulators. It remains unclear why previous studies, using the classical ammonium pre-pulse method, found no evidence for HVCN1-dependent flux (e.g. complete block of pH recovery when NHE was inactivated in $CO_2/HCO_3^-$-free superfusates), but the present findings mandate a re-evaluation of pH control in the heart. A role for HVCN1 may be important in the context of ischaemia and reperfusion, as well as cardiac hypertrophy.

Evidence points to the current being $H^+$-selective, Nernstian and highly temperature sensitive: the hallmarks of HVCN1 channels. Pharmacological responses were also consistent with this type of channel. Possibly the most compelling test for the homeostatic significance of a novel transport mechanism is to demonstrate that its blockade evokes a meaningful change to the intracellular concentration of the carried ions. Under pharmacological NHE1 blockade, HVCN1 inhibition produced a rapid and profound acidification, interpreted to indicate considerable $H^+$ traffic at resting pHi in paced cells. pHi decreased by $\sim$1 unit in 5 min, equivalent to an acidifying flux of $\sim$8 mmol $l^{-1}$ $min^{-1}$, if we take an average intrinsic buffering capacity over this pH range to be $\sim$40 mmol/(l cytosol)/pH (Leem et al., 1999). The source of this acid is unclear, but metabolism is one of the few processes capable of supplying such a sustained magnitude of acid. In well-perfused myocytes, glycolytic lactic acid production is unlikely because oxygen supply is ample. The other major metabolic source of acid is mitochondrial $CO_2$ production, which the authors postulate supplies the $H^+$ ions that are ultimately extruded by HVCN1 channels.

The magnitude of $CO_2$ production in the heart is well known. Myocardial $O_2$ consumption is 8−13 ml/100 g of tissue/min (Hoffman & Buckberg, 2014). Taking Avogadro's law, the heart's respiratory quotient ($CO_2$ production/$O_2$ consumption) of 0.8, and that 100 g of myocardial tissue contains 69 ml of water (Vinnakota & Bassingthwaighte, 2004), the resting $CO_2$ production rate is $\sim$6 mmol $l^{-1}$ $min^{-1}$. Since the electrically paced isolated myocytes under study were not mechanically loaded, their respiratory rate may be lower than this whole-tissue estimate. Nonetheless, the flux of $H^+$ ions carried by HVCN1 channels is similar to mitochondrial $CO_2$ production. This is significant, because it would indicate that virtually all $CO_2$ production is trafficked across the surface membrane as $H^+$ and $HCO_3^-$ ions, rather than as $CO_2$ gas. This is a striking observation because it represents a paradigm shift in our thinking of $CO_2$ handling. So far, it was firmly believed that essentially all mitochondrially produced $CO_2$ crosses the membrane as $CO_2$ gas. Even though $CO_2$ is hydrated in the cytoplasm by carbonic anhydrase enzymes, there will always remain a residual concentration of the gaseous form. According to the canonical model, mitochondrial $CO_2$ is vented across membranes in its gaseous form, rather than as its dissociated ions, as these are considered to be several orders of magnitude less permeable than uncharged molecules. The new proposal, which states that HVCN1 current (alongside a high-capacity $HCO_3^-$ permeability) carries the bulk of mitochondrial $CO_2$ output, implies that the membrane must have restricted capacity to conduct $CO_2$ gas, despite a favourable concentration gradient. In other words, if cardiomyocytes rely on HVCN1 channels to remove $CO_2$, their gas permeability is much lower than previously measured (Arias-Hidalgo et al., 2017). If HVCN1 channels are essential to overcome previously unrecognized gas permeability limitations, the findings are of major importance to our understanding of membrane biology. Moreover, to vent $CO_2$ as its hydration products, it is essential to match $H^+$ current with $HCO_3^-$; the identity of the latter process remains to be confirmed, with a possible role for $Cl^-/HCO_3^-$ exchangers.

The discovery of HVCN1 current in canine myocytes is an important contribution to the field. Its implications on our understanding of $CO_2$ permeability, as well as parallel $HCO_3^-$ conductances, are considerable and these need to be re-evaluated. Further studies will certainly

look into the role of HVCN1 channels in myocytes of other species, including humans and their role in heart diseases.

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

## Additional information

### Competing interests

None declared.

### Author contributions

Both authors have read and approved the final version of the manuscript submitted for publication. All persons designated as authors qualify for authorship, and all those who qualify for authorship are listed.

### Funding

British Heart Foundation (BHF): Pawel Swietach, RG/15/9/31534.

### Keywords

acid-base balance, cardiomyocyte, mitochondria, voltage-gated channel

## Supporting information

Additional supporting information can be found online in the Supporting Information section at the end of the HTML view of the article. Supporting information files available:

**Peer Review History**

