## [Peer Review History · The Journal of Physiology]

Channelling protons out of the heart

Pawel Swietach and Sanda Despa

DOI: 10.1113/JP283250

Corresponding author(s): Pawel Swietach (pawel.swietach@dpag.ox.ac.uk)

The following individual(s) involved in review of this submission have agreed to reveal their identity: Hong-Sheng Wang (Referee #1)

Review Timeline:

Submission Date:	04-Mar-2022
Editorial Decision:	19-Apr-2022
Resubmission Received:	23-Apr-2022
Editorial Decision:	28-Apr-2022
Revision Received:	28-Apr-2022
Accepted:	28-Apr-2022

Senior Editor: Bjorn Knollmann

Reviewing Editor: Eleonora Grandi

Transaction Report:

Dear Professor Swietach,

Re: JP-P-2022-283061 "Channelling protons out of the heart" by Pawel Swietach and Sanda Despa

Thank you for submitting your manuscript to The Journal of Physiology. Please accept our apologies for the delay in providing you with an editorial decision on your submission.

Your Perspective article has been assessed by a Reviewing Editor and by 1 Referee and the reports are copied below.

Please let your co-author know of the following editorial decision as quickly as possible.

As you will see, in its current form, the manuscript is not acceptable for publication in The Journal of Physiology. In comments to me, the Reviewing Editor expressed interest in the potential of this study, but much work still needs to be done in order to satisfactorily address the concerns raised in the reports.

In view of this interest, I would like to offer you the opportunity to carry out all of the changes requested in full, and to resubmit a new manuscript using the "Submit Special Case Resubmission for JP-P-2022-283061..." on your homepage.

We cannot, of course, guarantee ultimate acceptance at this stage as the revisions required are substantial. However, we encourage you to consider the requested changes and resubmit your work to us if you are able to complete or address all changes.

A new manuscript would be renumbered and redated, but the original referees would be consulted wherever possible. An additional referee's opinion could be sought, if the Reviewing Editor felt it necessary. A full response to each of the reports should be uploaded with a new version.

I hope that the points raised in the reports will be helpful to you.

Yours sincerely,

Bjorn Knollmann
Senior Editor
The Journal of Physiology

EDITOR COMMENTS

Reviewing Editor:

The perspective should be revised to highlight the importance and novelty of the study, so as to provide more balance.

Senior Editor:

The reviewer identified serious factual errors in the article. I concur. Unless appropriately revised, we will not be able to publish the piece.

REFEREE COMMENTS

Referee #1:

The manuscript is riddled with major factual errors. Also, the authors do not discuss the novelty and importance of the new findings (discovery of a powerful previously unknown Na⁺-independent mechanism of H⁺ extrusion in cardiac myocytes), nor do they discuss how it might relate to major outstanding questions about cardiac acid-base homeostasis in vivo. For example: Does the presence of HVCN1 in cardiac myocytes explain why NHE1 can be knocked out or inhibited with little or no effect on normal cardiac performance? Is there reciprocal regulation of NHE1 and HVCN1 (i.e., of Na-dependent and Na-independent acid extrusion)? As one example, when NHE1 is inhibited by hypoxia, as shown in Dr. Swietach's recent study and proposed as an ATP-sparing mechanism, is HVCN1 activated? Are there conditions in which reciprocal regulation might

be beneficial in order to increase or to reduce Na-loading? One would typically want to see such new possibilities discussed in a Perspectives article. Instead, the authors focus on the amount of acid observed in the cell culture studies and present an unexplained mathematical model and Figure that proposes to show that pH in perfused cardiac myocytes can be maintained in the absence of carbonic anhydrase activity. It is difficult to see how any of this is relevant to the findings of the Focus article or gives any insights into what kind of research these new findings might lead to in the future. Our specific comments are:

1. In the first paragraph, the authors write: "That said, the depolarization needed to drive H⁺ current ultimately requires ATP, so the energetic burden is not quite eliminated." This sentence does not follow well from the preceding sentence, but is probably referring to our assertion that the concerted activities of HVCN1, Cl/HCO₃ exchange, and Cl channel activity is an energetically efficient means of disposing of CO₂ hydration products. Of course, it is true that maintaining the transmembrane ionic gradient to support membrane excitation requires expenditure of energy, and it is also true that the electrogenic flux of any ion across the membrane would "influence the action potential"; however, activating the H channel does not require extra ATP. If H extrusion were carried out by Na/H exchange, one would have to dispose of Na via the Na pump (with additional expenditure of ATP). Also, as noted in the Focus article, extrusion of H⁺ would have a similar effect on the membrane potential as K⁺ extrusion (i.e., repolarization), so it may have a K-sparing effect (requiring less Na,K-ATPase activity). The authors of the Perspectives article state that "H⁺ removal through HVCN1 would strongly influence the action potential" but this has not been established. It would be worth proposing that this question deserves study, rather than claiming to know what would occur before doing the experiment. It is possible that K channel activity adjusts to the level of electrogenic H extrusion, but this remains to be determined.

2. In the second paragraph, the authors write: "Dialysing cells with a substantially acidic internal solution (pH_i=6.5) evoked an outward current of ~1 pA/pF at 0 mV under near-physiological extracellular pH." However, it is not the absolute pH value that matters, it is the transmembrane difference in pH. Activation of the Hv1 currents is not dependent upon the "substantially acidic internal solution". For example, at an internal pH of 7.5, Hv1 is also activated upon depolarization, but activation is just shifted to a different voltage. Also, as noted in the Focus article, the channel is heavily regulated, and its activity is likely much more robust in vivo than indicated in the experiments utilizing quiescent myocytes at room temperature in the highly specialized buffers needed to isolate H⁺ currents.

3. In the third paragraph, the authors briefly note that the characteristics of the channel match those of HVCN1, but then write that: "In the case of NHE1, inhibition has only a small effect on pH_i because this transporter is strategically activated by acidic cytoplasm, which avoids futile pumping of acid. In contrast, HVCN1 inhibition produced a rapid and profound acidification, interpreted to indicate considerable H⁺ traffic at rest." This is absurd and totally misrepresents the data. Blockade of either NHE1 or HVCN1 alone had only a small effect on pH_i, indicating that the other mechanism was sufficient to maintain pH_i in beating myocytes in culture. Please look at figure 5 G, H and I. It is the inhibition of both mechanisms together that caused a significant acidification. When Hv1 is blocked in beating myocytes, inhibition of NHE1 causes a rapid drop in pH_i, indicating that it is critical for pH_i regulation starting at normal pH_i. The same result was obtained when NHE1 was inhibited and then followed by HVCN1 inhibition. NHE1 is regulated by many mechanisms and can operate effectively at high pH_i; it is activated by acidic pH_i but it does not require acidic pH_i in order to be active. Furthermore, HVCN1 is also "strategically" activated by acidic cytoplasm, which also avoids futile pumping of acid, but it too is heavily regulated. Also, the myocytes were not resting. We made it abundantly clear both in the manuscript and in our responses to the repeated reviews that these were beating cells. Beating cells generate much more CO₂ than quiescent cells.

4. Also in the third paragraph (after only two sentences devoted to discussing the data documenting that HVCN1 is expressed and active in myocytes), the authors move on to a very lengthy and flawed discussion of the amount of acid produced and its sources. When estimating the amount of acid produced in beating myocytes the authors cited a paper by Leem et al, 1999 and said that the buffering power over the range that we studied (pH 7.2 - 6.2) was ~40 mM. However, this is the average buffering power over that range. The value that is relevant is the amount of additional buffered acid that accumulates as pH drops from 7.2 (where it is already ~27 mM) to 6.2 (where it rises to ~49 mM). The difference indicates that about 22 mM of additional acid accumulates. We cited two different papers (from the Vaughan-Jones and Wakabayashi labs) that each estimated ~26 mM in this range. Also, Referee 2 (of our paper), after suggesting in earlier reviews that the acid accumulation would be ~50 mM, acknowledged that in HEPES buffer the value would be half that (25 mM). If the value of 22 or 26 mM is used (either one is reasonable) to correct the value for CO₂ production estimated by the authors of the Perspectives article, then you get a value similar to the value that we estimated, which is fully in line with experimentally derived values for O₂ utilization and a respiratory quotient of 1.0, which is what it would be in media containing glucose (not 0.8 as suggested by the authors). This argument has been hashed over ad nauseum in the repeated reviews of our manuscript and would be of little interest to most readers. The simple fact is that calculations and models are valid ONLY when they are based on and consistent with real data. We present real data that prove the existence and importance of HVCN1 in cardiac myocytes. The Perspectives writers may wish this were not the case, but it is. Their wistful comment "Just when we thought that cardiomyocyte pH control had been fully characterized..." reveals their disappointment with the new

reality.

5. The final paragraph starts with the comment that "The authors postulate that mitochondrial CO₂ hydrates to cytoplasmic H⁺ and HCO₃⁻ ions and that the former is expelled through HVCN1". Actually, we did not have to postulate that CO₂ is efficiently hydrated as it exits the mitochondria; we cited data from a PNAS paper by Dr. Swietach (PMID: 23431149) that had already shown that (see their Fig. 2B). In their abstract they state that "Inhibition of extramitochondrial CA activity acidified the matrix" (of the mitochondria), "impaired cardiac energetics", and "reduced contractility." It is clear from those data that CO₂ is efficiently hydrated as it exits the mitochondria and this clearly indicates the need for extrusion of the hydration products (H⁺ via HVCN1 and NHE), along with residual CO₂, in order to maintain cardiac energetics and contractility.

The authors go on to state that "If the entire CO₂ output were routed through HVCN1 channels as H⁺ ions, it would imply that CO₂ cannot cross membranes, which is, of course, implausible. Indeed, if gases could not pass membranes, neither would O₂ needed for CO₂ production." Of course, this is absurd and we did not suggest it. In discussing the PNAS paper mentioned above and an earlier paper by some of the same authors (PMID: 20008827), which indicated about a 1:10 ratio of labeled CO₂ (from ¹³C pyruvate) to HCO₃⁻ (derived from hydration of labeled CO₂), we noted that the rapid decline of both labeled species indicated the elimination of newly formed HCO₃⁻ and residual CO₂. We have no doubt that CO₂ can diffuse across the sarcolemma, but the data from the Swietach and Schroeder labs cited above indicate that CO₂ is efficiently hydrated as it exits the mitochondria. So, the cell must remove both HCO₃⁻ and residual CO₂. Regarding O₂ permeability through membranes, we speculated (based on appropriate references) that the high cholesterol content of t-tubule membranes could allow O₂ entry from the t-tubular fluid, which exchanges rapidly with extracellular fluid during mechanical activity.

We are puzzled by two additional points in the final paragraph and figure legend. First, the authors state that "Although CO₂ hydrates to H and HCO₃⁻, its gaseous form can exit at a rate matching production." OK, so what? It may be true that CO₂ can exit entirely by diffusion, but the papers from the Swietach and Schroeder labs mentioned above indicate that it is largely hydrated as it exits the mitochondria and that this is necessary to maintain cardiac energetics and contractility, so it is necessary to dispose of both the hydration products and the residual CO₂. Second, the authors state that "A channel-mediated pathway would not make a meaningful contribution to CO₂ venting unless it is coupled to an equal HCO₃⁻ co-flux (OK, we agree with this, both hydration products would have to be extruded, we talked about that, and we know that myocytes have very powerful transport mechanisms for both H and HCO₃⁻ extrusion) and the membrane became impermeable to CO₂". We do not agree with this; why would H and HCO₃⁻ extrusion via transport mechanisms require that the sarcolemma be impermeable to CO₂? This is ridiculous.

They go on to say that "the HCO₃⁻ conductance that balances HVCN1 current is unclear, and the source of acid --- is unlikely to be mitochondrial". Again, this is absurd. Cl/HCO₃⁻ exchangers are expressed at very high levels in cardiac myocytes and have extraordinarily high turnover numbers. Loss of function of a single allele of the AE3 gene in humans causes heart disease (PMID: 29167417), so it is clearly important, as discussed in our manuscript. The idea that the source of acid is unlikely to be mitochondrial leaves us speechless. Hydration of CO₂ derived from mitochondrial metabolism is by far the greatest source of acid in biological tissues and Dr. Swietach's landmark PNAS paper cited above and discussed in our paper clearly indicates that robust hydration of CO₂ occurs as it exits the mitochondria.

Review of Perspectives Article

Is the Perspectives article factually accurate NO.

Comments for the Author

The manuscript is riddled with major factual errors. Also, the authors do not discuss the novelty and importance of the new findings (discovery of a powerful previously unknown Na^+ -independent mechanism of H^+ extrusion in cardiac myocytes), nor do they discuss how it might relate to major outstanding questions about cardiac acid-base homeostasis in vivo. For example: Does the presence of HVCN1 in cardiac myocytes explain why NHE1 can be knocked out or inhibited with little or no effect on normal cardiac performance? Is there reciprocal regulation of NHE1 and HVCN1 (i.e., of Na-dependent and Na-independent acid extrusion)? As one example, when NHE1 is inhibited by hypoxia, as shown in Dr. Swietach's recent study and proposed as an ATP-sparing mechanism, is HVCN1 activated? Are there conditions in which reciprocal regulation might be beneficial in order to increase or to reduce Na-loading? One would typically want to see such new possibilities discussed in a Perspectives article. Instead, the authors focus on the amount of acid observed in the cell culture studies and present an unexplained mathematical model and Figure that proposes to show that pH in perfused cardiac myocytes can be maintained in the absence of carbonic anhydrase activity. It is difficult to see how any of this is relevant to the findings of the Focus article or gives any insights into what kind of research these new findings might lead to in the future. Our specific comments are:

1. In the first paragraph, the authors write: *"That said, the depolarization needed to drive H^+ current ultimately requires ATP, so the energetic burden is not quite eliminated."* This sentence does not follow well from the preceding sentence, but is probably referring to our assertion that the concerted activities of HVCN1, Cl/HCO_3 exchange, and Cl channel activity is an energetically efficient means of disposing of CO_2 hydration products. Of course, it is true that maintaining the transmembrane ionic gradient to support membrane excitation requires expenditure of energy, and it is also true that the electrogenic flux of any ion across the membrane would "influence the action potential"; however, activating the H channel does not require **extra** ATP. If H extrusion were carried out by Na/H exchange, one would have to dispose of Na via the Na pump (with additional expenditure of ATP). Also, as noted in the Focus article, extrusion of H^+ would have a similar effect on the membrane potential as K^+ extrusion (i.e., repolarization), so it may have a K -sparing effect (requiring less Na,K -ATPase activity). The authors of the Perspectives article state that " H^+ removal through HVCN1 would strongly influence the action potential" but this has not been established. It would be worth proposing that this question deserves study, rather than claiming to know what would occur before doing the experiment. It is possible that K channel activity adjusts to the level of electrogenic H extrusion, but this remains to be determined.

2. In the second paragraph, the authors write: *“Dialysing cells with a substantially acidic internal solution (pHi=6.5) evoked an outward current of ~1 pA/pF at 0 mV under near-physiological extracellular pH.”* However, it is not the absolute pH value that matters, it is the transmembrane difference in pH. Activation of the Hv1 currents is not dependent upon the “substantially acidic internal solution”. For example, at an internal pH of 7.5, Hv1 is also activated upon depolarization, but activation is just shifted to a different voltage. Also, as noted in the Focus article, the channel is heavily regulated, and its activity is likely much more robust in vivo than indicated in the experiments utilizing quiescent myocytes at room temperature in the highly specialized buffers needed to isolate H⁺ currents.

3. In the third paragraph, the authors briefly note that the characteristics of the channel match those of HVCN1, but then write that: *“In the case of NHE1, inhibition has only a small effect on pHi because this transporter is strategically activated by acidic cytoplasm, which avoids futile pumping of acid. In contrast, HVCN1 inhibition produced a rapid and profound acidification, interpreted to indicate considerable H⁺ traffic at rest.”* This is absurd and totally misrepresents the data. Blockade of **either** NHE1 or HVCN1 alone had only a small effect on pHi, indicating that the other mechanism was sufficient to maintain pHi in beating myocytes in culture. Please look at figure 5 G, H and I. it is the inhibition of **both** mechanisms together that caused a significant acidification. When Hv1 is blocked in beating myocytes, inhibition of NHE1 causes a rapid drop in pHi, indicating that it is critical for pHi regulation starting at normal pHi. The same result was obtained when NHE1 was inhibited and then followed by HVCN1 inhibition. NHE1 is regulated by many mechanisms and can operate effectively at high pHi; it is activated by acidic pHi but it does not require acidic pHi in order to be active. Furthermore, HVCN1 is also “strategically” activated by acidic cytoplasm, which also avoids futile pumping of acid, but it too is heavily regulated. Also, the myocytes were not resting. We made it abundantly clear both in the manuscript and in our responses to the repeated reviews that these were beating cells. Beating cells generate much more CO₂ than quiescent cells.

4. Also in the third paragraph (after only two sentences devoted to discussing the data documenting that HVCN1 is expressed and active in myocytes), the authors move on to a very lengthy and flawed discussion of the amount of acid produced and its sources. When estimating the amount of acid produced in beating myocytes the authors cited a paper by Leem et al, 1999 and said that the buffering power over the range that we studied (pH 7.2 – 6.2) was ~40 mM. However, this is the average buffering power over that range. The value that is relevant is the amount of **additional** buffered acid that accumulates as pH drops from 7.2 (where it is already ~27 mM) to 6.2 (where it rises to ~49 mM). The difference indicates that about 22 mM of additional acid accumulates. We cited two different papers (from the Vaughan-Jones and Wakabayashi labs) that each estimated ~26 mM in this range. Also, Referee 2 (of our paper), after suggesting in earlier reviews that the acid accumulation would be ~50 mM, acknowledged that in HEPES buffer the value would be half that (25 mM). If the value of 22 or 26 mM is used (either one is reasonable) to correct the value for CO₂ production estimated by the authors of

the Perspectives article, then you get a value similar to the value that we estimated, which is fully in line with experimentally derived values for O₂ utilization and a respiratory quotient of 1.0, which is what it would be in media containing glucose (not 0.8 as suggested by the authors). This argument has been hashed over ad nauseum in the repeated reviews of our manuscript and would be of little interest to most readers. The simple fact is that calculations and models are valid ONLY when they are based on and consistent with real data. We present real data that prove the existence and importance of HVCN1 in cardiac myocytes. The Perspectives writers may wish this were not the case, but it is. Their wistful comment "*Just when we thought that cardiomyocyte pH control had been fully characterized...*" reveals their disappointment with the new reality.

5. The final paragraph starts with the comment that "*The authors postulate that mitochondrial CO₂ hydrates to cytoplasmic H⁺ and HCO₃⁻ ions and that the former is expelled through HVCN1*". Actually, we did not have to postulate that CO₂ is efficiently hydrated as it exits the mitochondria; we cited data from a PNAS paper by Dr. Swietach (PMID: 23431149) that had already shown that (see their Fig. 2B). In their abstract they state that "*Inhibition of extramitochondrial CA activity acidified the matrix*" (of the mitochondria), "*impaired cardiac energetics*", and "*reduced contractility*." It is clear from those data that CO₂ is efficiently hydrated as it exits the mitochondria and this clearly indicates the need for extrusion of the hydration products (H⁺ via HVCN1 and NHE), along with residual CO₂, in order to maintain cardiac energetics and contractility.

The authors go on to state that "*If the entire CO₂ output were routed through HVCN1 channels as H⁺ ions, it would imply that CO₂ cannot cross membranes, which is, of course, implausible. Indeed, if gases could not pass membranes, neither would O₂ needed for CO₂ production.*" Of course, this is absurd and we did not suggest it. In discussing the PNAS paper mentioned above and an earlier paper by some of the same authors (PMID: 20008827), which indicated about a 1:10 ratio of labeled CO₂ (from ¹³C pyruvate) to HCO₃ (derived from hydration of labeled CO₂), we noted that the rapid decline of both labeled species indicated the elimination of newly formed HCO₃ and residual CO₂. We have no doubt that CO₂ can diffuse across the sarcolemma, but the data from the Swietach and Schroeder labs cited above indicate that CO₂ is efficiently hydrated as it exits the mitochondria. So, the cell must remove both HCO₃ and residual CO₂. Regarding O₂ permeability through membranes, we speculated (based on appropriate references) that the high cholesterol content of t-tubule membranes could allow O₂ entry from the t-tubular fluid, which exchanges rapidly with extracellular fluid during mechanical activity.

We are puzzled by two additional points in the final paragraph and figure legend. **First**, the authors state that "*Although CO₂ hydrates to H and HCO₃, its gaseous form can exit at a rate matching production.*" OK, so what? It may be true that CO₂ can exit entirely by diffusion, but the papers from the Swietach and Schroeder labs mentioned above indicate that it is largely hydrated as it exits the mitochondria and that this is necessary to maintain cardiac energetics

and contractility, so it is necessary to dispose of both the hydration products and the residual CO₂. **Second**, the authors state that “A channel-mediated pathway would not make a meaningful contribution to CO₂ venting **unless it is coupled to an equal HCO₃ co-flux** (OK, we agree with this, both hydration products would have to be extruded, we talked about that, and we know that myocytes have very powerful transport mechanisms for both H and HCO₃ extrusion) **and the membrane became impermeable to CO₂**”. We do not agree with this; why would H and HCO₃ extrusion via transport mechanisms require that the sarcolemma be impermeable to CO₂? This is ridiculous.

They go on to say that “the HCO₃ conductance that balances HVCN1 current is unclear, and the source of acid --- is unlikely to be mitochondrial”. Again, this is absurd. Cl/HCO₃ exchangers are expressed at very high levels in cardiac myocytes and have extraordinarily high turnover numbers. Loss of function of a single allele of the AE3 gene in humans causes heart disease (PMID: 29167417), so it is clearly important, as discussed in our manuscript. The idea that the source of acid is unlikely to be mitochondrial leaves us speechless. Hydration of CO₂ derived from mitochondrial metabolism is by far the greatest source of acid in biological tissues and Dr. Swietach’s landmark PNAS paper cited above and discussed in our paper clearly indicates that robust hydration of CO₂ occurs as it exits the mitochondria.

Comments for the Editor

This is a particularly poor Perspectives article that does nothing to capture the importance of the finding that HVCN1 serves as a major pHi regulator in heart, nor does it discuss possibilities for new avenues of research that are opened up by these findings. As noted above, it contains major factual errors, and statements that are confusing, irrelevant, or simply wrong. As just one example, the authors stated: *“If the entire CO₂ output were routed through HVCN1 channels as H⁺ ions, it would imply that CO₂ cannot cross membranes, which is, of course, implausible. Indeed, if gases could not pass membranes, neither would O₂ needed for CO₂ production.”* This is a classical Straw Man Fallacy. They put words into our mouths that we never said or suggested (or would ever suggest), and then refute those words. The concluding paragraph and Figure present an unexplained mathematical model that seems to be referring to isolated non-beating myocytes in culture that has no relationship to our findings.

In the review of an earlier version of this manuscript, Referee 1 stated that: *“The work is both important and novel as it focuses on a Na-independent mechanism for removing an intracellular acid load”* and Referee 2 stated that: *“This is a thought-provoking manuscript that would be a paradigm shift in our understanding of pH regulation in the heart.”* We agree with those assessments about the potential importance of our study. It is disturbing to see a Perspectives article that does nothing to inform the reader about the potential impact of these new findings on our understanding of cardiac pH regulation.

We (Hong-Sheng Wang, Tom DeCoursey, Gary Shull, and Jianyong Ma) think that this Perspectives article is deeply biased and mis-states several fundamental aspects of our paper. Because it is so deeply flawed, we do not think that it merits publication. As an alternative, we would like to suggest the Cross-Talk format, in which each of the differing views can be presented and openly debated.

Thank you for collating the feedback on our Perspective. We are sorry to hear the article was not acceptable. We wish to submit a revision. We feel that certain core arguments presented in our opinion should remain, as they put the work in perspective of fluxes, which is an essential test of any new homeostatic mechanism. We hope we can reach a compromise with Referee 1, presumably one of the authors of the original manuscript that the perspective is on.

Our responses to the specific comments are given in red below.

Referee #1:

The manuscript is riddled with major factual errors. Also, the authors do not discuss the novelty and importance of the new findings (discovery of a powerful previously unknown Na⁺-independent mechanism of H⁺ extrusion in cardiac myocytes), nor do they discuss how it might relate to major outstanding questions about cardiac acid-base homeostasis in vivo. For example: Does the presence of HVCN1 in cardiac myocytes explain why NHE1 can be knocked out or inhibited with little or no effect on normal cardiac performance? Is there reciprocal regulation of NHE1 and HVCN1 (i.e., of Na-dependent and Na-independent acid extrusion)? As one example, when NHE1 is inhibited by hypoxia, as shown in Dr. Swietach's recent study and proposed as an ATP-sparing mechanism, is HVCN1 activated? Are there conditions in which reciprocal regulation might be beneficial in order to increase or to reduce Na-loading? One would typically want to see such new possibilities discussed in a Perspectives article. Instead, the authors focus on the amount of acid observed in the cell culture studies and present an unexplained mathematical model and Figure that proposes to show that pH in perfused cardiac myocytes can be maintained in the absence of carbonic anhydrase activity. It is difficult to see how any of this is relevant to the findings of the Focus article or gives any insights into what kind of research these new findings might lead to in the future.

We recognise there are concerns about how we interpret your important findings. We propose to re-phase some of our points, tone down our critique, and provide a more balanced perspective that highlights your achievements.

The revised Perspective has three sections: (i) an introduction to the field and quantification of HVCN1 current relative to other transporters, (ii) a summary of the author's findings, and (iii) putting the authors results in the context of mitochondrial CO₂ production. We attempted to summarise the key findings and properties, but in the interest of space, we cannot replicate more detail.

We recognise that your findings are a major paradigm shift in the field. As part of community-driven verification, all homeostatic mechanisms require a careful analysis of fluxes to test (i) if the proposed system is feasible (e.g. thermodynamically) and (ii) how the new system compares with established mechanisms. The original manuscript did not present a comprehensive flux analysis, nor did it compare fluxes against the existing framework, therefore we felt a Perspective article on these matters would be useful, particularly by showing a mathematical model. Historically, studies of pH regulation relied on the tandem of experiment and mathematics, and we are merely following a standard process. We trust the authors have confidence in their experimental measurements, thus applying these to a model should not evoke any concerns. The model we present is published, and equations are available in the original reference. This has been the basis for all our CA-related work. However, we are content to remove the figure and model if this serves to re-balance the article. Instead of a model, we explain the implications of HVCN1 in words.

1. In the first paragraph, the authors write: "That said, the depolarization needed to drive H⁺ current ultimately requires ATP, so the energetic burden is not quite eliminated." This sentence does not follow well from the preceding sentence, but is probably referring to our assertion that the concerted activities of HVCN1, Cl/HCO₃ exchange, and Cl channel activity is an energetically efficient means of disposing of CO₂ hydration products. Of course, it is true that maintaining the transmembrane ionic gradient to support membrane excitation requires expenditure of energy, and it is also true that the electrogenic flux of any ion across the membrane would "influence the action potential"; however, activating the H channel does not require extra ATP. If H extrusion were carried out by Na/H exchange, one would have to dispose of Na via the Na pump (with additional expenditure of ATP). Also, as noted in the Focus article, extrusion of H⁺ would have a similar effect on the membrane potential as K⁺ extrusion (i.e., repolarization), so it may have a K-sparing effect (requiring less Na,K-ATPase activity). The authors of the Perspectives article state that "H⁺ removal through HVCN1 would strongly influence the action potential" but this has not been established. It would be worth proposing that this question deserves study, rather than claiming to know what would occur before doing the experiment. It is possible that K channel activity adjusts to the level of electrogenic H extrusion, but this remains to be determined.

Both sides concur that any H⁺ current would ultimately require ATP, but the issue is how this compares against NHE etc. We revised this part of the Perspective and removed the problematic sentences.

2. In the second paragraph, the authors write: "Dialysing cells with a substantially acidic internal solution (pH_i=6.5) evoked an outward current of ~1 pA/pF at 0 mV under near-physiological extracellular pH." However, it is not the absolute pH value that matters, it is the transmembrane difference in pH. Activation of the Hv1 currents is not dependent upon the "substantially acidic internal solution". For example, at an internal pH of 7.5, Hv1 is also activated upon depolarization, but activation is just shifted to a different voltage. Also, as noted in the Focus article, the channel is heavily regulated, and its activity is likely much more robust in vivo than indicated in the experiments utilizing quiescent myocytes at room temperature in the highly specialized buffers needed to isolate H⁺ currents.

We agree; but the purpose of this calculation was to measure flux in familiar units. It is really very important that a transporter's pH regulatory power is judged using standard units of H⁺ flux, thus we are performing the calculation to show how much flux is produced by HVCN1, and compare this to NHE. We believe a Perspective's role is to compare your results with that of others, and doing so requires like-for-like comparisons. We conclude that the HCVN mechanism is comparable to NHE1 – surely a welcome statement.

3. In the third paragraph, the authors briefly note that the characteristics of the channel match those of HVCN1, but then write that: "In the case of NHE1, inhibition has only a small effect on pH_i because this transporter is strategically activated by acidic cytoplasm, which avoids futile pumping of acid. In contrast, HVCN1 inhibition produced a rapid and profound acidification, interpreted to indicate considerable H⁺ traffic at rest." This is absurd and totally misrepresents the data. Blockade of either NHE1 or HVCN1 alone had only a small effect on pH_i, indicating that the other mechanism was sufficient to maintain pH_i in beating myocytes in culture. Please look at figure 5 G, H and I. It is the inhibition of both mechanisms together that caused a significant acidification. When Hv1 is blocked in beating myocytes, inhibition of NHE1 causes a rapid drop in pH_i, indicating that it is critical for pH_i regulation starting at normal pH_i. The same result was obtained when NHE1 was inhibited and then followed by HVCN1 inhibition. NHE1 is regulated by many mechanisms and can operate effectively at high pH_i; it is activated by acidic pH_i but it does not require acidic pH_i in order to be active.

Furthermore, HVCN1 is also "strategically" activated by acidic cytoplasm, which also avoids futile pumping of acid, but it too is heavily regulated. Also, the myocytes were not resting. We made it abundantly clear both in the manuscript and in our responses to the repeated reviews that these were beating cells. Beating cells generate much more CO₂ than quiescent cells.

We changed "resting" to "resting pHi" to emphasise that the pH change is from a physiological pHi. It has been established that NHE1 inhibition has only a subtle effect on pHi from a resting level because most acid production can be vented across the membrane as CO₂ or lactic acid. This can be seen in pHi-NHE flux curves, which typically drop well below 1 mM/min near resting pH. The fact that HVCN1 inhibition produces a rapid and profound drop in pHi must be highlighted as an entirely novel observation, something we have not seen before with any other drug.

4. Also in the third paragraph (after only two sentences devoted to discussing the data documenting that HVCN1 is expressed and active in myocytes), the authors move on to a very lengthy and flawed discussion of the amount of acid produced and its sources.

As reviewers, we felt that the most striking feature of HVCN1 was the magnitude of flux, particularly at resting pHi (effect of drug). This evoked the natural question – what is the source of this acid? This aspect was not studied in detail by the paper, so we felt it was appropriate to consider the system as a whole.

When estimating the amount of acid produced in beating myocytes the authors cited a paper by Leem et al, 1999 and said that the buffering power over the range that we studied (pH 7.2 - 6.2) was ~40 mM. However, this is the average buffering power over that range. The value that is relevant is the amount of additional buffered acid that accumulates as pH drops from 7.2 (where it is already ~27 mM) to 6.2 (where it rises to ~49 mM). The difference indicates that about 22 mM of additional acid accumulates. We cited two different papers (from the Vaughan-Jones and Wakabayashi labs) that each estimated ~26 mM in this range.

We can assure the authors that there is no flaw here. The pH range over which the observation was made was 7.2 to 6.2, over which buffering capacity rises from 27 to 49. Taking a mean of 40 is a reasonable approximation (the precise number is 38). Thus, we can calculate the amount of acid produced by multiplying buffering capacity by pH change.

Also, Referee 2 (of our paper), after suggesting in earlier reviews that the acid accumulation would be ~50 mM, acknowledged that in HEPES buffer the value would be half that (25 mM). If the value of 22 or 26 mM is used (either one is reasonable) to correct the value for CO₂ production estimated by the authors of the Perspectives article, then you get a value similar to the value that we estimated, which is fully in line with experimentally derived values for O₂ utilization and a respiratory quotient of 1.0, which is what it would be in media containing glucose (not 0.8 as suggested by the authors).

There is no flaw here, as our calculation was for the myocardium where RQ is 0.8 (please bear in mind the myocardium respire fatty acids). This flux is useful as it indicates the magnitude of CO₂ production rate we may expect in myocytes, i.e. a useful reference point. If the authors propose that mitochondrial CO₂ production is the source of HVCN1 H⁺ flux, surely it is appropriate to put numbers to these. The referee should not feel offended by our efforts to compare literature values to their findings. Indeed, the purpose of a Perspective is to put your work in perspective. The calculations presented confirm that HVCN1 flux is large, of the order of several mM/min; we are putting your observations in perspective.

This argument has been hashed over ad nauseum in the repeated reviews of our manuscript and would be of little interest to most readers. The simple fact is that calculations and models are valid ONLY when they are based on and consistent with real data. We present real data that prove the existence and importance of HVCN1 in cardiac myocytes. The Perspectives writers may wish this were not the case, but it is.

In the tradition of JP, we feel it is important to consider numbers. Your elegant paper provides a measure of flux, which we assign a number to. We then compare this against CO₂ production in the myocardium – to put things in perspective. The readers of JP will want to see numerical analyses, rather than qualitative descriptions. Our perspective firmly confirms the size of HVCN1 flux. We are unclear why this confirmation has attracted criticism.

Their wistful comment "Just when we thought that cardiomyocyte pH control had been fully characterized..." reveals their disappointment with the new reality.

We are not disappointed – on the contrary, we are pleased a pH-related article will appear in JP. We revised this sentence to ensure it is not misunderstood.

5. The final paragraph starts with the comment that "The authors postulate that mitochondrial CO₂ hydrates to cytoplasmic H⁺ and HCO₃⁻ ions and that the former is expelled through HVCN1". Actually, we did not have to postulate that CO₂ is efficiently hydrated as it exits the mitochondria; we cited data from a PNAS paper by Dr. Swietach (PMID: 23431149) that had already shown that (see their Fig. 2B). In their abstract they state that "Inhibition of extramitochondrial CA activity acidified the matrix" (of the mitochondria), "impaired cardiac energetics", and "reduced contractility." It is clear from those data that CO₂ is efficiently hydrated as it exits the mitochondria and this clearly indicates the need for extrusion of the hydration products (H⁺ via HVCN1 and NHE), along with residual CO₂, in order to maintain cardiac energetics and contractility.

Correct; CO₂ is hydrated at equilibrium to H⁺ and HCO₃⁻. The same is true for weak acids, such as acetic acid, which dissociates to H⁺ and acetate; however, trafficking across the membrane takes the form of acetic acid because uncharged molecules diffuse faster than ions. We and others subscribe to the notion that CO₂ hydrates, but efflux across membranes takes the form of CO₂ predominantly. There is a wealth of information in support of this. However, the new paper is a paradigm-shift as it suggests that CO₂ leaves cells as H⁺ and HCO₃⁻, instead. This inference must be highlighted explicitly and tested, as it is truly a paradigm-shift.

Thank you for quoting our work, but please bear in mind that our paper proposes that CO₂ leaves cells as CO₂ gas, which is in direct contradiction to the present work.

The authors go on to state that "If the entire CO₂ output were routed through HVCN1 channels as H⁺ ions, it would imply that CO₂ cannot cross membranes, which is, of course, implausible. Indeed, if gases could not pass membranes, neither would O₂ needed for CO₂ production." Of course, this is absurd and we did not suggest it.

To clarify, we are merely analysing the author's data. CO₂ production in the myocardium is ~6 mM/min. Based on the effect of inhibition, HVCN1 carries a flux of 8 mM/min. We believe the authors agree with these numbers. Thus, the activity of HVCN1 carries the entire bulk of H⁺-equivalent of CO₂ production. From simple sums, this leaves no space for CO₂ to traffic as CO₂ gas (of course, a single molecule of CO₂ cannot leave a cell twice). To paraphrase, these calculations mean that the entire CO₂ output crosses the membrane as H⁺. This, in turn, can only take place if CO₂ permeability is much lower than previously believed. In proposing that CO₂ is removed as H⁺

the authors implicitly suggest that the membrane hinders CO₂ transit in the form of gas. We believe this is important to explain in the Perspective, as it is part of the paradigm-shift.

In discussing the PNAS paper mentioned above and an earlier paper by some of the same authors (PMID: 20008827), which indicated about a 1:10 ratio of labeled CO₂ (from ¹³C pyruvate) to HCO₃ (derived from hydration of labeled CO₂), we noted that the rapid decline of both labeled species indicated the elimination of newly formed HCO₃ and residual CO₂. We have no doubt that CO₂ can diffuse across the sarcolemma, but the data from the Swietach and Schroeder labs cited above indicate that CO₂ is efficiently hydrated as it exits the mitochondria. So, the cell must remove both HCO₃ and residual CO₂. Regarding O₂ permeability through membranes, we speculated (based on appropriate references) that the high cholesterol content of t-tubule membranes could allow O₂ entry from the t-tubular fluid, which exchanges rapidly with extracellular fluid during mechanical activity.

CO₂ is certainly hydrated, but it is not removed completely. The current consensus states that 90% of CO₂ exists as HCO₃ at physiological pH, but essentially all transmembrane traffic takes the form of uncharged gas. This is the same situation as with acetic acid – most of which exists in the dissociated form, but crosses the membrane as the uncharged acid. Schroeder et al cannot be used to support the present study, as your conclusions are contradictory.

We are puzzled by two additional points in the final paragraph and figure legend. First, the authors state that "Although CO₂ hydrates to H and HCO₃, its gaseous form can exit at a rate matching production." OK, so what?

If CO₂ exits in gaseous form at the same rate as production, then there is no need for H⁺ transport. If this held true, it would mean no role for HVCN1. We think this statement is very important to clarify, but we have now rephrased it to put it in context.

It may be true that CO₂ can exit entirely by diffusion, but the papers from the Swietach and Schroeder labs mentioned above indicate that it is largely hydrated as it exits the mitochondria and that this is necessary to maintain cardiac energetics and contractility, so it is necessary to dispose of both the hydration products and the residual CO₂.

This is a misunderstanding of our work – CO₂ is hydrated but the residual CO₂ is membrane-permeable, and there is no need for transporting CO₂ as H⁺ and HCO₃⁻. By analogy, cytoplasmic weak acids such as acetic acid exist mainly as acetate and H⁺, but they cross membranes as acetic acid. If H⁺ channels were needed to carry acetic acid away, it would indicate an insufficiency of acetic acid permeability (something that would be a radical departure from the consensus).

Second, the authors state that "A channel-mediated pathway would not make a meaningful contribution to CO₂ venting unless it is coupled to an equal HCO₃ co-flux (OK, we agree with this, both hydration products would have to be extruded, we talked about that, and we know that myocytes have very powerful transport mechanisms for both H and HCO₃ extrusion) and the membrane became impermeable to CO₂". We do not agree with this; why would H and HCO₃ extrusion via transport mechanisms require that the sarcolemma be impermeable to CO₂? This is ridiculous.

This is not ridiculous, but its simple addition. Let's say that

Total CO₂ production = total CO₂ efflux

And that

$$\text{Total CO}_2 \text{ efflux} = (\text{Efflux as CO}_2 \text{ gas}) + (\text{Efflux as H}^+ \text{ and HCO}_3^-)$$

Let's rename the last equation as

$$T = C + H$$

We agreed that T is 6 mM/min (CO₂ production rate in myocardium) and the authors now show that H is 8 mM. This means that C cannot be greater than zero (in fact, its -2 but lets ignore this discrepancy for now). Essentially, the authors' conclusion is that CO₂ does not exit the cell as CO₂, but rather as H⁺ (and HCO₃⁻). If there is no capacity for CO₂ exit from cells, despite a gradient, the only conclusion is that the cell membrane is (relatively) impermeable to CO₂. If this is not the case, the original manuscript misled the reviewers by saying the purpose of HVCN1 is to carry the components of CO₂ hydration (presumably because these cannot cross as the gas). Please clarify.

They go on to say that "the HCO₃ conductance that balances HVCN1 current is unclear, and the source of acid --- is unlikely to be mitochondrial". Again, this is absurd. Cl/HCO₃ exchangers are expressed at very high levels in cardiac myocytes and have extraordinarily high turnover numbers. Loss of function of a single allele of the AE3 gene in humans causes heart disease (PMID: 29167417), so it is clearly important, as discussed in our manuscript. The idea that the source of acid is unlikely to be mitochondrial leaves us speechless. Hydration of CO₂ derived from mitochondrial metabolism is by far the greatest source of acid in biological tissues and Dr. Swietach's landmark PNAS paper cited above and discussed in our paper clearly indicates that robust hydration of CO₂ occurs as it exits the mitochondria.

We removed these lines and rephrased the observations in a more toned way. In retrospect, that is how we should have presented the Perspective.

Dear Dr Swietach,

Re: JP-P-2022-283250X "Channelling protons out of the heart" by Pawel Swietach and Sanda Despa

Thank you for submitting your invited Perspectives article to The Journal of Physiology. It has been assessed by a Reviewing Editor and the author of the focus paper.

Minor alterations have been requested.

The reports are copied at the end of this email. Please address all of the points and incorporate all requested revisions.

NEW POLICY: In order to improve the transparency of its peer review process The Journal of Physiology publishes online as supporting information the peer review history of all articles accepted for publication. Readers will have access to decision letters, including all Editors' comments and referee reports, for each version of the manuscript and any author responses to peer review comments. Referees can decide whether or not they wish to be named on the peer review history document.

I hope you will find the comments helpful and have no difficulty in revising your article within 7 days.

To submit the revised version use the links in Author Tasks Link Not Available.

Please ensure that the article is a Word File with no more than 5 references, including the focus paper.

Thank you for your contribution to the Journal.

Yours sincerely,

Bjorn Knollmann
Senior Editor
The Journal of Physiology

EDITOR COMMENTS

Reviewing Editor:

The authors have significantly revised the commentary and provided a more balanced perspective. One remaining statement about the contribution of HVCN1 inhibition to acidification needs correction.

Senior Editor:

I concur with the reviewing editor.

REFeree COMMENTS:

Referee #1:

We appreciate the authors' willingness to consider most of our suggestions, and feel that the revised Perspective is more balanced and objective, and its tone more reasonable.

There is one remaining misstatement that we hope that the authors can correct. Paragraph 2, line 5, the authors stated that "HVCN1 inhibition produced a rapid and profound acidification". It was the blockade of both Hv1 and NHE1, not Hv1 alone, that produced significant acidification. Hv1 blockade alone only cause minor change in pH (see figure 5G and H). These results are important because they point to the roles of both HVCN1 and NHE in proton extrusion. The misstatement may confuse the readers and we hope can be corrected.

We clarified in paragraph 2 that the effect of Hv1 inhibition was under NHE blockade.

Dear Dr Swietach,

Re: JP-P-2022-283250XR1 "Channelling protons out of the heart" by Pawel Swietach and Sanda Despa

I am pleased to tell you that your invited Perspective article has been accepted for publication in The Journal of Physiology.

NEW POLICY: In order to improve the transparency of its peer review process The Journal of Physiology publishes online as supporting information the peer review history of all articles accepted for publication. Readers will have access to decision letters, including all Editors' comments and referee reports, for each version of the manuscript and any author responses to peer review comments. Referees can decide whether or not they wish to be named on the peer review history document.

The last Word version of the paper submitted will be used by the Production Editors to prepare your proof. When this is ready you will receive an email containing a link to Wiley's Online Proofing System. The proof should be checked and corrected as quickly as possible.

All queries at proof stage should be sent to tjp@wiley.com

Thank you very much for your contribution to The Journal of Physiology.

Yours sincerely,

Bjorn Knollmann
Senior Editor
The Journal of Physiology

Reviewing Editor Comments:

Thanks for clarifying the statement.

Senior Editor:

Excellent work!

Reviewer Comments:

Referee #1:

Thank you. We have no further comments.

2nd Confidential Review**28-Apr-2022**